# Multiple Immunostainings with Different Epitope Retrievals—The FOLGAS Protocol

**DOI:** 10.3390/ijms23010223

**Published:** 2021-12-25

**Authors:** Anna von Schoenfeld, Peter Bronsert, Michael Poc, Andrew Fuller, Andrew Filby, Stefan Kraft, Konrad Kurowski, Kristin Sörensen, Julia Huber, Jens Pfeiffer, Michele Proietti, Verena Stehl, Martin Werner, Maximilian Seidl

**Affiliations:** 1Institute for Surgical Pathology, Medical Center—University of Freiburg and Faculty of Medicine, University of Freiburg, 79106 Freiburg, Germany; a.willemer@hotmail.de (A.V.S.); peter.bronsert@uniklinik-freiburg.de (P.B.); michael.poc@t-online.de (M.P.); konrad.kurowski@uniklinik-freiburg.de (K.K.); Kristin.werner@uniklinik-freiburg.de (K.S.); Julia.huber@uniklinik-freiburg.de (J.H.); martin.werner@uniklinik-freiburg.de (M.W.); 2Clinic for Internal Medicine, Berlin Jewish Hospital, 13347 Berlin, Germany; 3Tumorbank Comprehensive Cancer Center Freiburg, University of Freiburg, 79106 Freiburg, Germany; 4Core Facility for Histopathology and Digital Pathology, Medical Center—University of Freiburg and Faculty of Medicine, University of Freiburg, 79106 Freiburg, Germany; 5Klinik Gut St. Moritz, 7500 St. Moritz, Switzerland; 6Flow Cytometry Core Facility, Innovation, Methodology and Application Research Theme, Leech Building, Faculty of Medical Sciences, Newcastle University, Newcastle upon Tyne NE1 7RU, UK; andrew.fuller@newcastle.ac.uk (A.F.); Andrew.Filby@newcastle.ac.uk (A.F.); 7Center for Dermatopathology, 79106 Freiburg, Germany; stefkraft@hotmail.com; 8Department of Oto-Rhino-Laryngology, Medical Center—University of Freiburg and Faculty of Medicine, University of Freiburg, 79106 Freiburg, Germany; pfeiffer@hnoamtheater.de; 9Praxis HNO am Theater, 79098 Freiburg, Germany; 10Center for Chronic Immunodeficiency, Institute for Immunodeficiency, Medical Center—University of Freiburg and Faculty of Medicine, University of Freiburg, 79106 Freiburg, Germany; proietti.michele@mh-hannover.de; 11Klinik für Rheumatologie und Immunologie, Hannover Medical School (MHH), 30625 Hannover, Germany; 12Institute of Pathology, Heinrich Heine University and University Hospital of Duesseldorf, 40225 Duesseldorf, Germany; Verena.Stehl@med.uni-duesseldorf.de

**Keywords:** immunofluorescence, immunohistochemistry, CyTOF, multiple staining, re-fixation, formalin, retrieval, crossreactivity, hydrophobic masking

## Abstract

We describe a sequential multistaining protocol for immunohistochemistry, immunofluorescence and CyTOF imaging for formalin-fixed, paraffin-embedded specimens (FFPE) in the formalin gas-phase (FOLGAS), enabling sequential multistaining, independent from the primary and secondary antibodies and retrieval. Histomorphologic details are preserved, and crossreactivity and loss of signal intensity are not detectable. Combined with a DAB-based hydrophobic masking of metal-labeled primary antibodies, FOLGAS allows the extended use of CyTOF imaging in FFPE sections.

## 1. Introduction

Immunofluorescence- (IHC) and immunohistochemistry- (IF) based staining represent standard methods for the detection of target molecule localization. Most of the formalin fixed and paraffin embedded (FFPE) specimens need an epitope retrieval to allow the binding of antibodies on proteins denatured during the FFPE-workup [1]. Multiple staining allows the analyses of protein co-localization of cells (e.g., lymphocytes) or cellular compartments (e.g., cell membrane) within one tissue section. The gain of informative value and economic benefit (especially in the case of small sample size) is higher compared to serial single staining. The major limitation for multi IHC and IF staining is the required use of antibodies derived from different host species in order to avoid crossreactivity. Commonly used protocols to circumvent these limitations are (a) antibody stripping [2,3,4], (b) denaturation of the first reaction complex by microwaves [5,6,7], (c) tyramide signal amplification [8,9] or (d) clearing/bleaching of the precedent chromogen / fluorophore followed by image capture and sequential image overlay [4,10,11]. Of note, these protocols need special buffers [3,12], additional techniques [10,13,14] and/or large quantities of hard disc storage space [4,10]. A few years ago, imaging mass spectrometry (CyTOF) was introduced as an elegant way to study multiple proteins in tissue sections through laser-mediated spatially controlled pyrolysis of tissue sections and antibodies labelled with metals, which are detected by time of flight mass spectrometry [15,16]. Despite the huge advantage of visualizing multiple markers in one session, reports using FFPE sections with comparable marker numbers to cryosections are sparse [17] due to the limitation to combine different epitope retrievals on one FFPE section.

We developed a protocol applicable for every laboratory working with IHC, IF and/or CyTOF based on the formalin gas phase (FOLGAS) for the re-fixation of tissue slides and the denaturation of the previously formed antigen-antibody complex. Hereby, primary antibodies from the same and/or different host species together with various antigen retrievals can be used on one single tissue section. Consumables and reagents are used in the daily practice of IHC and IF. When metal-labelled antibodies for CyTOF imaging are coated with DAB, which is simply introduced by a secondary antibody coupled with horseradish peroxidase, the advantages of FOLGAS are transferrable, enabling the combination of different retrieval methods in CyTOF imaging, too.

## 2. Results

### 2.1. FOLGAS Enables the Combination of Different Epitope Retrievals with Maintained Tissue Quality throughout the Different Retrievals

In the first approach, we wanted to identify FoxP3 (nuclear stain of mainly regulatory T-cells) and KLRG1 positive (mainly natural killer cells; all antibodies and retrievals are listed in Table 1) cells. KLRG1 staining needs proteinase K (ProtK) retrieval whereas FoxP3 does not work with ProtK pretreatment. For the first staining experiments, we performed KLRG1 staining directly followed by FoxP3 staining and vice versa, with the retrievals as necessary prior to primary antibody incubation. Due to strong tissue and signal degradation (Figure 1A,B), a fixation step before the second epitope retrieval was implemented in order to prevent tissue degradation during the second retrieval step after the first staining. As we saw tissue degeneration during short-phase 4% aqueous formalin fixation of cryosections, we could not exclude maceration of the FFPE tissue section in the liquid phase of formalin after the IHC procedures. Therefore, we decided to perform the re-fixation in the gas phase over 35% formalin on the slide. After the first staining procedure, slides—still moistened after washing off unbound dyes—were stored in 50 mL tubes with 35% formalin in aqueous solution only in the conus (~5 mL) at 37 °C overnight, cap closed, without direct contact between the tissue section and the formalin. The evaporated formaldehyde allowed us to maintain tissue stability on the slide throughout the staining sequences. The double staining (described above) for KLRG1 and FoxP3 was successfully conducted (Figure 1C). Subsequently, staining procedures were performed up to a triple-staining, using 20% formalin in aqueous solution without methanol, the last step being an antibody cocktail (examples are given in Figure 1D–F, secondary antibodies, chromogens, fluorophores and epitope retrievals listed in Table 2,). Counterstaining was performed using hematoxylin for IHC, and DAPI 1:1000 for IF after the final staining. Coverslips were mounted with aqueous mounting medium or fluorescent mounting medium (flow chart given in Figure 2).

### 2.2. Subsequent Retrieval Steps Provide Maintained Signal Visibility but May Alter Signal Intensity

Concerning signal intensity, quantity and subcellular localization, we tested whether the multi-staining reveals similar results compared to the respective single-staining. We focused on the structural allocation of membrane and nuclear antigens known from macrophages (CD163, CD68, RBPJ; Figure 3) for IF and nuclear antigens known from B-cells and regulatory T-cells (BCL6, PAX5, FoxP3; Figure 4). All slides were evaluated semiquantitatively by two independent pathologists for cellular localization (membranous, cytoplasmic or nuclear), amount of stained cells (percentage), staining intensity (negative, weak, moderate or strong) and topographic localization of the staining (germinal center, mantle zone, interfollicular area, epithelium). Considering the amount of stained cells and staining intensity, an H-Score was calculated (staining intensity × amount of stained cells). Epitope retrievals were necessary and performed prior to incubating the primary antibodies. All primary antibodies were followed by secondary antibodies +/− fluorophores/chromogens as listed in Table 2.

For the immunofluorescent stainings, the sum of H-scores was lower for the first of the triple staining, equal for the second and higher for the third of the triple staining, compared to the matched single staining (Figure 3). Yet, the weaker signal was traceable throughout the slide with comparable numbers of positive cells (Appendix A).

For the immunohistochemical stainings, the sum of H-scores was lower for the first of the triple staining, slightly lower for the second and similar for the third of the triple staining, compared to the matched single staining (Figure 4, Appendix A).

### 2.3. When Combined with a DAB Based Masking of the Metal Labeling, the FOLGAS Protocol Could Be Successfully Transferred in a Pilot Cytometry by Time of Flight (CyTOF) Imaging Experiment

We postulated that FOLGAS may also be useful to overcome the limitations of only one retrieval in CyTOF imaging (Hyperion Imaging Mass Cytometry, Fluidigm, London, UK), rendering it more suitable for its use in FFPE tissue sections. However, first analyses did not reveal any signal. As the primary antibodies were still detectable with fluorophore-coupled secondary antibodies, we postulated that the metal labeling may be lost during the subsequent retrieval step and sought a way to locally immobilize it, e.g., by coating it with a hydrophobic product. Therefore, the metal-labelled primary antibodies were incubated with an HRP-coupled secondary antibody, followed by a DAB-based chromogen reaction. With this covalently bound brown product, we successfully hydrophobically cloaked the metal labelling, which could still be mobilized through the laser-mediated pyrolysis for further CyTOF imaging, resulting in a successful detection (Figure 5).

## 3. Discussion

The use of antibodies represents the gold standard technique for spatially resolved protein detection and finds its application in histopathological diagnostics and scientific research. For the establishment of multiple staining protocols, curse and blessing is the use of primary antibodies, derived from different species. FOLGAS directly tackles this obstacle and paves the way for antibody species independent multiple staining protocols, for research and histopathological diagnostic analyses. FOLGAS represents an easy to use protocol for multiple stainings, as the only modification of the original protocols for multiple staining consists in the interposed fixation step in the formaldehyde containing gas phase over 20% formalin. It is applicable in laboratories equipped with a basic infrastructure for FFPE based immunohistochemistry/-fluorescence or in laboratories with slide based staining techniques and access to formalin. Additional needed reagents are freely available chromogens or fluorophores. The tissue structure is well maintained due to the refixation process. The crossreactivity is prevented due to the denaturation of unsaturated antigen binding sites. Whether this absence of crossreactivity is owed to the chemical denaturation during the formaldehyde gas refixation, or the heat-mediated denaturation during (subsequent) epitope retrievals prior to the next antibody incubation, remains speculative. Absent staining with primary and secondary antibodies after the refixation step (data not shown) substantiates the former. This theory is also supported by Wang et al. using formaldehyde vapored from paraformaldehyde powder heated at 80 °C together with dried slides [18], however, without having excluded heat mediated denaturation in their data communication.

As the secondary antibodies used for IF are directly labeled or coupled with the fluorophore via streptavidin/biotin linkage and fluorescent signals are lost upon stripping [12], we postulate the fluorophores being linked to the tissue via the antibodies and not due to direct formalin crosslinks, in accordance with others denaturing slide bound antibodies by microwave heating [6]. Lan et al. provided clear evidence of ameliorated epitope retrieval during microwave heating in a citric acid buffer at pH6, rendering visible initially masked epitopes of FFPE-sections, with the limitation of only one possible membrane antigen staining [5]. During our analyses, we stained up to two different membrane proteins after different epitope retrievals (Figure 1, Figure 3 and Figure 5). Of note, the CD68 signal was getting weaker when being stained at first, independent of the fluorophore used (data not shown), which was the reason for CD68 being stained as second marker in the triple IF. Background staining was not an issue with the antibodies used in this study, but cannot be excluded in other settings. Of note, the establishment of multiple stainings requires robust single staining and critical evaluation of every antibody.

In the IHC multistainings, strongly hydrophobic birefringent precipitations of 1–5 µm may occur. Since they have not been observed in the IF stainings and do not look like formalin pigments, we assume that they resulted from a reaction or co-precipitation of IHC chromogens. We could provoke similar precipitations when pipetting paraffin saturated xylene on an objective plate, either after drying or, more importantly, directly followed by rinsing the paraffin saturated xylene droplet with 100% ethanol (Appendix A). Therefore, we would suggest that those precipitations are caused by incomplete deparaffinization.

Further, ProtK pretreatment may compromise previous staining steps, which is why we recommend staining the antigens requiring ProtK pretreatment first. The AP-based blue chromogen was not well suitable for the refixation, since its signal intensity diminished significantly. We therefore used it for the last staining step.

From a personal point of view, we would sum up by recommending the following steps if one plans to apply the FOLGAS protocol: (1) Each of the single stainings should work properly, including knowledge of the retrievals. To determine the optimal dilution, one could start with 5 µg/mL antibody concentration on the section; (2) Try different orders of the antibodies throughout the sequential stainings; (3) Chromogens or fluorophores prone to bleaching should be stained last.

Taken together, FOLGAS has the potential to facilitate the application of multiple IHC stainings. One of the first fields for the application of the FOLGAS protocol could be the immunohistological evaluation of immune checkpoint proteins within different tissue components. In short, immune checkpoint proteins control immune response in order to prevent autoimmunity. By adapting immune checkpoint proteins, cancer cells escape from the immune response and can continue with tumor cell proliferation and migration. Immune checkpoint proteins can be identified within the tumor and the tumor surrounding immune cells as e.g., cytotoxic CD8+ T-cells, dendritic cells and macrophages [19,20,21]. Therapies with immune checkpoint inhibitors may depend on the expression of Programmed death-ligand 1 (PD-L1) by tumor cells, tumor-associated immune cells, or both [22]. By establishing a tumor specific IHC panel (e.g., for lung adenocarcinomas: thyroid transcription factor-1 (TTF-1), CD68 and PD-L1), patients who can potentially benefit from immune checkpoint blockade therapy could be identified by using only one tissue slide, sparing tissue for further diagnostics.

Nevertheless, to introduce FOLGAS in routine diagnostics, the duration of the different protocol steps have to be optimized. For example, the dedicated FOLGAS protocol for the mentioned IHC lung adenocarcinoma panel would take three working days, which could be too long in comparison with only one working day for three single stainings on different sections.

A potential improvement could be the automation of the FOLGAS protocol, e.g., in autostainers by including the refixation step, rendering the FOLGAS protocol more comparable, faster and more reproducible.

## 4. Materials and Methods

For all experiments we used two µm thick formalin fixed paraffin embedded sections (FFPE) from palatine tonsils of immunologically healthy donors after positive ethical vote by local authorities (251/13_140389, Ethikkommission der Albert-Ludwigs-Universität Freiburg, Freiburg, Germany) and written informed consent, mounted on coated class objective plates. Deparaffinization was performed by immersion in xylene followed by ethanol with increasing percentages of deionized water for rehydration. Sections subjected to the optimal epitope retrieval (Table 1), followed by blocking either endogenous biotin or endogenous peroxidases prior to incubation with the primary antibody (Table 1). All primary antibodies were incubated for 60 min at room temperature. The dilution is displayed in Table 1 and Table 2. Afterwards, sections were incubated for 20 min at room temperature with a mixed polyclonal anti-mouse/anti-rabbit secondary antibody (Table 2) which was either biotinylated or directly coupled to a horseradish peroxidase (HRP) containing protein backbone (Table 2). The biotinylated secondary antibodies were used for streptavidine-linked alkaline phosphatase dependent chromogen reactions for IHC (red, green or blue; Table 2) or for streptavidin linked fluorophores for IF (excitation wavelengths of 390, 488 or 555 nm, respectively; Table 2). The HRP coupled secondary antibodies were used for a DAB based brown chromogen reaction in IHC (Table 2). Antibody-metal conjugations for imaging mass cytometry were performed as per the manufacturers protocol using a commercially available kit (cat no: SKU 201300; Hyperion Imaging Mass Cytometry and labeling kits, Fluidigm, London, UK). Conjugation success was determined by incubating the processed antibody with Iridium-imbued antibody capture beads (ABC bead, Thermo Fisher, Loughborough, UK, Cat no: A10497) and acquired on a fully calibrated Helios Mass Cytometry (Fluidigm) in suspension mode. The concentration of antibody pre and post conjugation was checked using a the Quibit protein kit (Thermo Fisher, cat no: Q33211). Antigen retrieval for Imaging Mass Cytometry (IMC) staining was performed as outlined in the FOLGAS method (Figure 2, Table 1). However, the final step required Iridium addition as per described [23]. Stained slides were then acquired on a fully calibrated Hyperion IMC system. The reader might notice a strong focus on immunological markers, which is merely owed to the interest of the authors in immunopathology.

## 5. Conclusions

The FOLGAS protocol is a widely usable multistaining method for immunohistochemistry, immunofluorescence and CyTOF imaging.

## Figures and Tables

**Figure 1 ijms-23-00223-f001:**
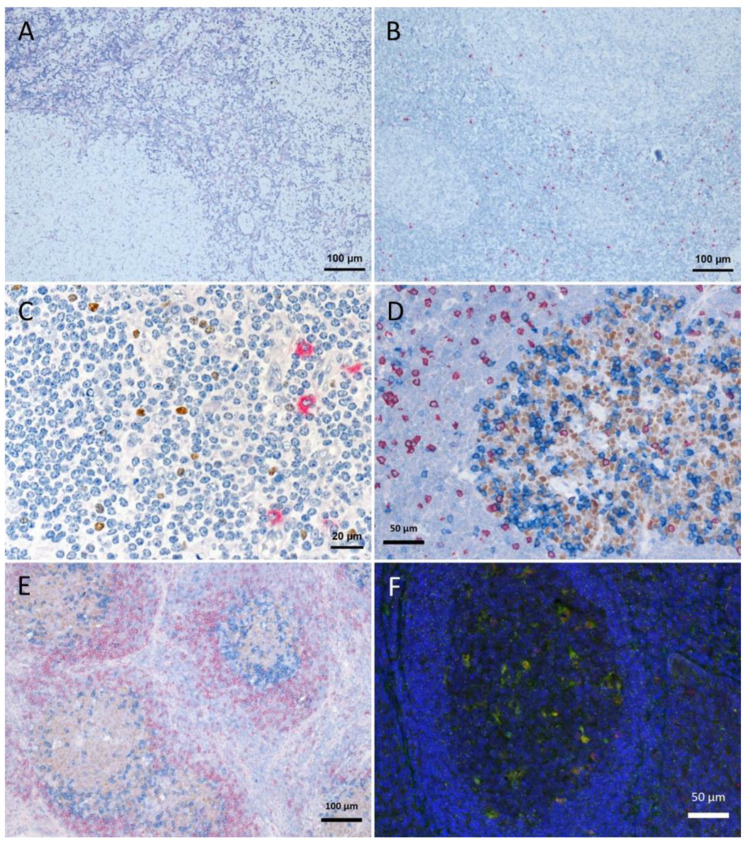
Tissue degradation and signal loss vs. preserved tissue and signal with refixation. (**A**) FoxP3 staining (brown) after 5 min retrieval in high pressure cooker in citric buffer pH 6, followed by KLRG1 (red) after proteinase K pretreatment and (**B**) vice versa (proteinase K followed by high pressure cooking) lead to strong tissue degradation. (**C**–**F**) show examples with refixation between the retrievals. (**C**) Protocols from (**B**) with interposed refixation step lead to preservation of tissue structure with maintained signals (KLRG1 red, FoxP3 brown). (**D**,**E**) Examples of triple staining with up to two retrievals: (**D**) human tonsil, CD8 (red, 2 min high pressure cooking) followed by Bcl6 (brown, 5 min high pressure cooking) together with PD1 (blue, cocktail with Bcl6); (**E**) human lymph node, IgD (red, without retrieval), followed by Bcl6/PD1 (brown and blue, respectively, after 5 min high pressure cooking). (**F**) immunofluorescence in human tonsil: LAMP1 (488 green, after 15 min TRIS pH 9 retrieval), ADA2 (555 red, after 20 min TRIS pH 6.1 retrieval), nuclear counterstain with DAPI (blue fluorescence).

**Figure 2 ijms-23-00223-f002:**
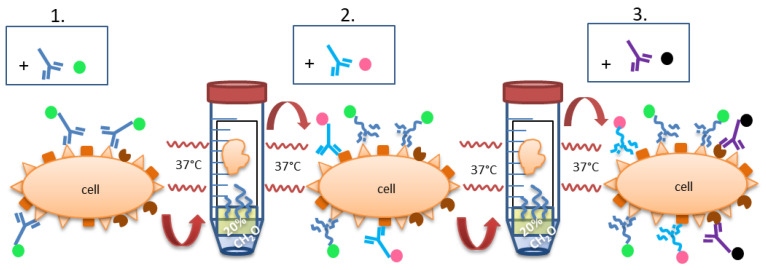
General staining technique: Tissue re-fixation in formalin gas phase. The color dots at the Fc part of the antibodies stand for direct labeling or labeling via secondary antibodies. The latter can be the same throughout the different staining steps because of the formaldehyde-mediated denaturation. (**1.**) IHC or IF staining is performed as usual, with either a directly labeled primary antibody or with primary + secondary antibody. After the last washing step, the still wet slide is brought into the gas phase over 20% formalin and incubated overnight at 37 °C, leading to re-fixation of the section, denaturation of the antibodies and their fixation on the slide. (**2.**) Second IHC or IF staining is performed as usual, followed by counterstaining (e.g., with hematoxylin or DAPI) and finalization or followed by the next round of re-fixation and staining (**3.**).

**Figure 3 ijms-23-00223-f003:**
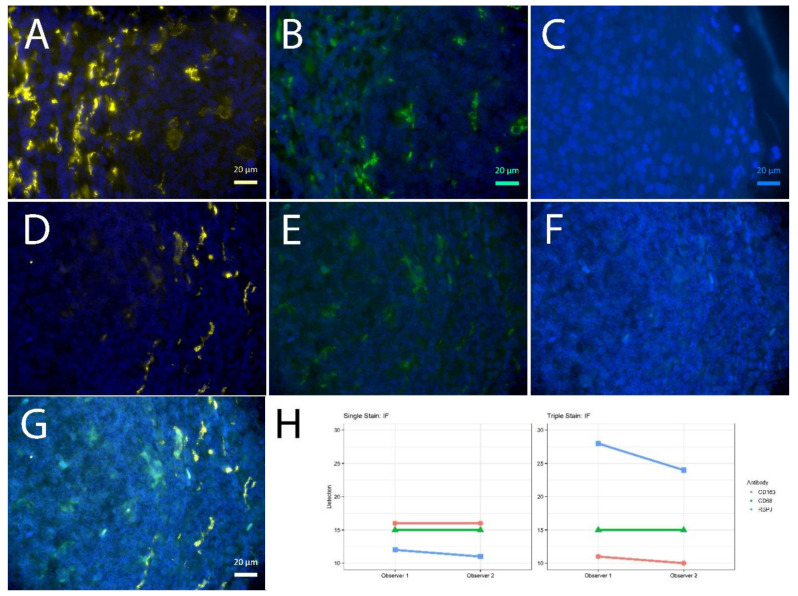
Single vs. triple stainings, immunofluorescence on human tonsil sections. (**A**–**C**) Single stainings: (**A**) CD163 555 gold, 2 min high pressure cooking citric buffer pH 6 (HPC), (**B**) CD68 488 green, 20 min TRIS pH 6.1, (**C**) RBPJ Aqua light blue, 2 min HPC. CD163 555 gold after 2 min HPC; CD68 488 green after pH 6.1, RBPJ Aqua light blue after 2 min HPC; DAPI as counterstain, respectively. (**D**–**F**) Triple staining according to the FOLGAS protocol, separated by fluorescent channels with DAPI as nuclear background dye; antigen retrievals equivalent to the corresponding single staining: (**D**) CD163 555 gold, (**G**) CD68 488 green, (**C**) RBPJ Aqua light blue. (**G**) Triple staining, merged. Magnifications indicated by bar. (**H**) Sum of H-scores of the respective stainings under single and triple staining conditions, from two independent observers.

**Figure 4 ijms-23-00223-f004:**
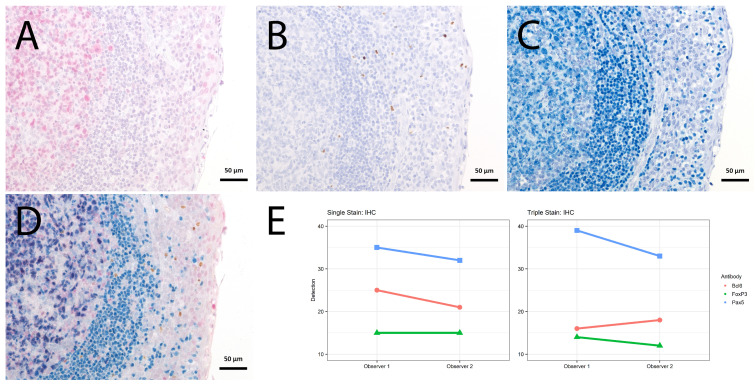
Single vs. triple stainings, immunohistochemistry on tonsil sections. (**A**–**C**) Single stainings: (**A**) Bcl6 red after 15 min TRIS pH 9, (**B**) FoxP3 brown after 20 min TRIS pH 6.1, (**C**) PAX5 blue after 20 min TRIS pH 6.1. (**D**) Human tonsil with triple staining IHC, according to the FOLGAS protocol: Bcl6 red, FoxP3 brown, PAX5 blue; antigen retrievals equivalent to the corresponding single staining. Magnifications indicated by bar. (**E**) Sum of H-scores of the respective stainings under single and triple staining conditions, from two independent observers.

**Figure 5 ijms-23-00223-f005:**
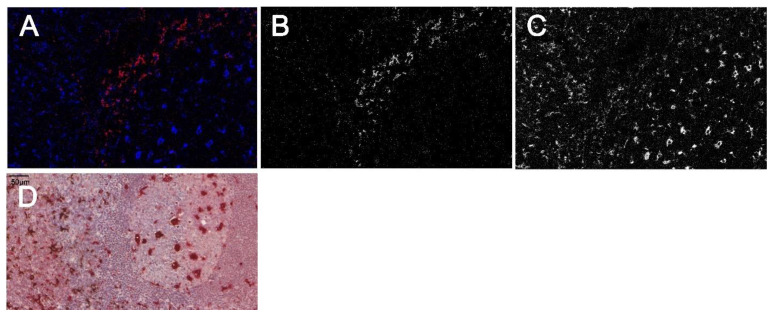
CyTOF based double staining with the modified FOLGAS protocol. (**A**–**C**) CyTOF ablation image: (**A**) merged image with CD163 in red (corresponding gray scale image in (**B**)) and CD68 in blue (corresponding gray scale image in (**C**)). (**D**) Double staining of the same slide prior to ablation. CD163 was stained first and developed/coated with a secondary antibody and DAB based brown color to shield the metal-labeled primary antibody. CD68 was stained second and labeled in red for better recognition. Magnification indicated by bar.

**Table 1 ijms-23-00223-t001:** Primary antibodies used, including retrievals. Abbreviations: RTU: Ready to use, HPC: high pressure cooking at 120 °C in citric buffer pH6.

Antibody Target	Clone	Species	Dilution	Provider	Retrieval
ADA2 (=CECR1)	Polyclonal	Rabbit	1:80	Prestige Sigma (HPA007888)	20 min TRIS pH 6.1
Bcl6	PG-B6p	Mouse	RTU	Dako (GA62561-2)	2 min HPC 15 min TRIS pH 9
CD163	EDHu-1	Mouse	1:500	BioRad (MCA1853)	2 min HPC
CD68	KP1	Mouse	1:1000	Biolegend (916104)	30 min TRIS pH 6.1
CD68	PG-M1	Mouse	RTU	Dako (GA61361-2)	20 min TRIS pH 6.1
CD8	C8/144B	Mouse	RTU	Dako (GA62361-2)	2 min HPC
FoxP3	236A/E7	Mouse	1:100	Thermo Fisher eBioscience (14-4777-82)	5 min HPC 20 min TRIS pH 6.1
KLRG1	13F12F2	Mouse	1:160	Gift from H. Pircher (hybridoma supernatant)	Prot K
LAMP1 (=CD107a)	eBioH4A3	Mouse	1:100	Thermo Fisher (14-1079-80)	15 min TRIS pH 9
PAX5	DAK-Pax5	Mouse	RTU	Dako (IR65061-2)	20 min TRIS pH 6.1
PD-1	Polyclonal	Goat	1:100	R&D Systems (AF1086)	5 min HPC
RBPJ	Polyclonal	Rabbit	1:50	Thermo Fisher (PA5-35187)	2 min HPC

**Table 2 ijms-23-00223-t002:** Secondary antibodies, chromogens, fluorophores and metals. Except DAPI, the fluorophores not directly coupled to antibodies are linked to streptavidin for binding of biotinylated antibodies. Abbreviations: RTU: ready to use, AP: alkaline phosphatase, HRP: horseradish peroxidase. Tb: Terbium, Er: Erbium.

Target Antibody	Chromogen/Fluorophore	Species/Enzyme	Dilution	Provider
Mouse + Rabbit	None, biotinylated	Goat	RTU	Dako (K5005)
Mouse	Alexa555	Donkey	1:200	Abcam (ab150106)
Rabbit	Alexa488	Goat	1:200	Life technologies (A-11008)
Mouse	None, biotinylated	Goat	1:200	Abcam (ab6788)
	Red	AP	-	Dako (K5005)
	Brown	HRP	-	Dako (K8010)
	Blue	AP	-	Abcam (ab178453)
	**Fluorophore**			
	Light blue		1:200	Interchim (Streptavidin—Fluoprobes 390A; FP-BM7700)
	Green		1:200	Thermo Fisher (Streptavidin AlexaFluor 488 Conjugate; S11223)
	Gold/orange		1:200	Thermo Fisher (Streptavidin AlexaFluor 555 Conjugate; S21381)
	Blue		-	Vectorlabs (Vectashield Antifade Mounting Medium with DAPI; H-1200-10)
	**Metal**			
	159Tb (conjugated to CD68 clone KP1)		-	Fluidigm (SKU 201300)
	170Er (conjugated to CD163 clone EDHu-1)		-	Fluidigm (SKU 201300)

## Data Availability

Aid with protocols is provided by the corresponding author on reasonable request.

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
