# Peer review of "Multiple Immunostainings with Different Epitope Retrievals—The FOLGAS Protocol"

_ijms, 2021, doi:10.3390/ijms23010223_

Round 1

Reviewer 1 Report

Dear Editor, thank you so much for inviting me to revise this manuscript.

This study addresses a current topic.

The manuscript is quite well written and organized. English could be improved.

Figures and tables are comprehensive and clear.

The introduction explains in a clear and coherent manner the background of this study.

We suggest the following modifications:

  • Methods and Statistical Analysis: nothing to add.
  • Discussion section: Very interesting and timely discussion. Of note, the authors should expand the Discussion section, including a more personal perspective to reflect on. For example, they could answer the following questions – in order to facilitate the understanding of this complex topic to readers: what potential does this study hold? What are the knowledge gaps and how do researchers tackle them? How do you see this area unfolding in the next 5 years? We think it would be extremely interesting for the readers.

However, we think the authors should be acknowledged for their work. In fact, they correctly addressed an important topic, the methods sound good and their discussion is well balanced.

One additional little flaw: the authors could better explain the limitations of their work, in the last part of the Discussion.

We believe this article is suitable for publication in the journal although some revisions are needed. The main strengths of this paper are that it addresses an interesting and very timely question and provides a clear answer, with some limitations.

We suggest a linguistic revision and the addition of some references for a matter of consistency. Moreover, the authors should better clarify some points.

Reviewer 2 Report

Authors successfully described a sequential multistaining protocol for immunohistochemistry, immunofluorescence and CyTOF imaging for formalin-fixed, paraffin-embedded specimens in the formalin gas-phase, enabling sequential multistaining, independent from the primary and secondary antibodies and retrieval. All the results are presented with great attention to detail. And thus, since this a protocol proposal, it would be very useful to add reference numbers of the used antibodies (Table 1 and 2) as well as incubation times (4. Material and Methods) for replication studies or as an introductory step for other laboratories. 

Round 2

Reviewer 1 Report

The authors modified the manuscript according to our suggestions.

We recommend Acceptance.